# Impaired Mitophagy in Neurons and Glial Cells during Aging and Age-Related Disorders

**DOI:** 10.3390/ijms221910251

**Published:** 2021-09-23

**Authors:** Vladimir Sukhorukov, Dmitry Voronkov, Tatiana Baranich, Natalia Mudzhiri, Alina Magnaeva, Sergey Illarioshkin

**Affiliations:** Research Center of Neurology, Department for Brain Research, 125367 Moscow, Russia; voronkovdm@gmail.com (D.V.); baranich_tatyana@mail.ru (T.B.); mudzhirinm@gmail.com (N.M.); alinamagnaeva03@gmail.com (A.M.); snillario@gmail.com (S.I.)

**Keywords:** mitophagy, aging, neurodegeneration, Parkinson’s disease, Alzheimer’s disease

## Abstract

Aging is associated with a decline in cognitive function, which can partly be explained by the accumulation of damage to the brain cells over time. Neurons and glia undergo morphological and ultrastructure changes during aging. Over the past several years, it has become evident that at the cellular level, various hallmarks of an aging brain are closely related to mitophagy. The importance of mitochondria quality and quantity control through mitophagy is highlighted by the contribution that defects in mitochondria–autophagy crosstalk make to aging and age-related diseases. In this review, we analyze some of the more recent findings regarding the study of brain aging and neurodegeneration in the context of mitophagy. We discuss the data on the dynamics of selective autophagy in neurons and glial cells during aging and in the course of neurodegeneration, focusing on three mechanisms of mitophagy: non-receptor-mediated mitophagy, receptor-mediated mitophagy, and transcellular mitophagy. We review the role of mitophagy in neuronal/glial homeostasis and in the molecular pathogenesis of neurodegenerative disorders, such as Parkinson’s disease, Alzheimer’s disease, and other disorders. Common mechanisms of aging and neurodegeneration that are related to different mitophagy pathways provide a number of promising targets for potential therapeutic agents.

## 1. Introduction

Aging is accompanied by a decline in cognitive function in a significant part of the population and is a major risk factor for the development of most neurodegenerative diseases, including Alzheimer’s disease (AD) and Parkinson’s disease (PD) [1]. Mechanisms of brain aging remain poorly understood, and studies aimed at the development of targeted therapy for neurodegenerative diseases are an important task for modern medicine.

Over the past few decades, it has been shown that aging processes in the brain are closely associated with mitochondrial dysfunction, resulting in oxidative stress and bioenergetic deficiency in various cells of the nervous tissue, which are extremely sensitive to energy deprivation [2,3]. Mitochondrial dysfunction is also a key pathological marker for neurodegenerative diseases. For example, it is a central player in sporadic and familial forms of PD, since dopaminergic neurons of the substantia nigra are especially vulnerable to energy deficiency due to their ability for autonomous activity, constant recirculation of synaptic vesicles, and developed axonal network [4]. Therefore, maintaining adequate energy levels through a functional pool of mitochondria is critical for neuron survival and function.

The main mechanism that prevents the development of mitochondrial dysfunction is mitophagy. Mitophagy is a complex multicomponent process that ensures control of the quality and quantity of mitochondria by eliminating damaged forms of these organelles via autophagy [5]. The importance of mitophagy for neurons may be explained by the significant role of the cytoplasmic renewal system for postmitotic cell populations. Maintaining the basal level of mitophagy is critical for ensuring the correct functioning of neurites since the bulk of mitochondria are localized in the distal parts of neuronal processes [5]. Mitophagy restricts the production of reactive oxygen species, prevents the accumulation of mitochondrial DNA (mtDNA) mutations and the decrease in ATP production, and blocks apoptotic signaling and the activation of inflammasomes [5,6]. It is the progressive decline in this type of selective macroautophagy throughout life that appears to lead to mitochondrial dysfunction and aging [6].

Many neurodegenerative diseases are characterized by the accumulation of neurotoxic protein aggregates resulting from mutations in the genes encoding for proteins that trigger mitophagy: PTEN-induced kinase 1 (PINK1), parkin, and protein deglycase DJ-1, among others [7]. Therefore, determination of the relationships between mitophagy markers and various parameters of neurodegenerative processes in PD, AD, and other age-related disorders seems to be highly promising from both a clinical and fundamental point of view.

## 2. Mitophagy in Neurons in Aging and Neurodegeneration

Dysfunctional mitochondria can influence the neuron aging process in two ways. Firstly, they are the main sources of reactive oxygen species, which contribute to cell damage. Secondly, due to a lack of histones and less efficient repair processes in the mitochondria, mitochondrial DNA is more susceptible to damage by reactive oxygen species, which makes it prone to the accumulation of a large number of mutations [8,9]. Thus, selective autophagy of mitochondria prevents the accumulation of their dysfunctional forms, which are an important factor in aging [7]. However, despite considerable evidence indicating the role of mitochondrial dysfunction in age-related involution and age-related diseases, determining the exact mechanisms of the effect of mitophagy impairments on aging processes still requires in-depth study.

It is assumed that mitochondrial fragments can be captured both non-selectively, by common mechanisms of macroautophagy, and selectively, with the involvement of receptor proteins and proteins of the outer membrane of mitochondria [10,11]. Consequently, mitophagy in neurons is based on three mechanisms: non-receptor-mediated mitophagy, receptor-mediated mitophagy (Figure 1), and transcellular mitophagy [12,13].

These mechanisms of mitochondrial degradation in neurons act and are regulated independently. The structure of a neuronal cell—hyperpolarization, and differences in axonal and somatic compartments—determines the features of mitophagy regulation in neurons. Mitochondria are predominantly located in the processes of neurons, while lysosomes are closer to the cell nucleus. The activation of mitophagy leads to a decrease in anterograde transport and an increase in retrograde transport, suggesting that, for mitophagy, damaged mitochondria are transported from the processes to the cell body [14]. In this regard, the mechanisms of degradation of damaged mitochondria by local mitophagy in neurites, or with the participation of retrograde transport of damaged organelles into the soma of a neuron, are discussed. Some studies have shown the degradation of axonal mitochondria along the PINK–parkin pathway [15]; however, the question of spatial regulation of mitophagy in neurons remains open [11,16].

The importance of mitophagy in the pathogenesis of neurodegenerative diseases is supported by the identified mutations of proteins involved in mitophagy—primarily PINK1 and parkin in Parkinson’s disease, optineurin and TBK1 in amyotrophic lateral sclerosis [17,18,19], and p62/SQSTM1 in frontotemporal dementia [20]. In addition to genetic disorders of the mitophagy pathway in a number of neurodegenerative diseases, including Alzheimer’s disease, the role of accumulation of damaged mitochondria in neurons is emphasized, as well as mitophagy disorders resulting from the interaction of components of the mitophagy pathway, for example, Drp1, with the accumulation pathological proteins, such as synuclein, tau protein, and beta-amyloid, in neurons [21,22,23].

### 2.1. PINK1–Parkin Pathway of Neuron Mitophagy (Non-Receptor-Mediated Mitophagy)

The mechanism of classical mitophagy (that is, mitophagy not mediated by receptors) is based on the induction of the mitochondrial membrane protein PTEN 1 serine–threonine kinase (PINK1) and parkin protein (PARK2), which is a cytosolic ubiquitin E3 ligase [12,13]. Thus, protein coding by *PINK1* (*PARK7*) and *PARK2* genes, which are associated with familial forms of PD, is a major factor in controlling mitochondrial quality. Known substrates for PINK1 include ubiquitin and the ubiquitin-homologous parkin domain, both of which are phosphorylated at a conserved serine residue (S65) [14]. The phosphorylation of these targets leads to the activation of parkin, followed by the formation of autophagosomes and the capture and degradation of damaged mitochondria [12,14]. The translocation of parkin from the cytosol into the outer mitochondrial membrane depends on PINK1 activity. Parkin, in turn, catalyzes ubiquitination and proteasome degradation of various proteins of the outer mitochondrial membrane, including dynamin-related protein 1 (Drp1), mitochondrial Rho (Miro), and mitofusins 1 and 2 (MFN1/2) [12,14]. This mechanism blocks the fusion of mitochondria, allowing for the isolation of damaged mitochondria and the initiation of the autophagy process through a system of adaptor proteins, including microtubule-associated protein 1A/1B-light chain 3 (LC3) and GABA type A receptor-associated protein (GABARAP) [12,14]. The scheme for classical non-receptor-mediated mitophagy is depicted in Figure 1.

Since PINK1 is the only known kinase that catalyzes ubiquitin phosphorylation, the detection of S65-phospho-ubiquitin can be used to assess PINK1 activity and is considered a biomarker of mitochondrial stress and autophagy [23,24,25]. Mitochondrial damage as a result of degradation by the protease presenilin-associated rhomboid-like protein (PARL) localized on the inner mitochondrial membrane leads to the accumulation of PINK1 [14]. Unlike classical sporadic PD, autopsy material obtained from patients with PINK1 or parkin mutations may not show Lewy bodies in substantia nigra neurons [26]. It is assumed that this is due to the participation of PINK1 and parkin in the long-term survival of dopaminergic neurons, and PINK1 and parkin defects lead to the rapid death of nigral neurons without the accumulation of pathological proteins, which is also confirmed by experiments with PINK1 knockdown.

#### 2.1.1. Mitophagy and Aging

In a transgenic line of mice, it was shown in vivo that the level of mitophagy decreased with age in the neurons of the dentate fascia of the hippocampus [27]. Aging models using *Drosophila melanogaster* and *Caenorhabditis elegans* showed an increase in life expectancy with the overexpression of *PARK2*, *PINK1*, *p62/SQSTM1*, *Drp1* [28,29,30]. In the study of Liang on elder mice (24 months), an ATG9-dependent decrease in the formation of autophagosomes in the mitochondria of cardiomyocytes as well as an increase in the content of parkin and the protein ubiquitination in the brain was revealed [31]. In addition to neurons, increased expression of parkin and NIX (BNIP3L) in brain vessels during aging was shown [32]. In aging *PARK2* knockout mice, the death of dopamine neurons was detected, and violation of microtubule stability in dopamine neurons was revealed, which emphasizes the role of parkin in the regulation of mitochondrial mobility [33,34]. Using cell models, it was also shown that PINK1 is necessary for the long-term survival of neurons, and that aging neurons are subject to apoptosis in its absence [35]. 

#### 2.1.2. Mitophagy and Neurodegeneration

It is known that genetic early onset forms of PD can be caused by mutations in genes *PARK2* (parkin), *PINK1*, and *DJ-1*, which encode proteins that are localized in mitochondria, and the loss of these proteins leads to an increased sensitivity to oxidative stress and impaired energy metabolism [36]. Another risk factor for hereditary PD is mutations in the glucocerebrosidase (*GBA*) gene, which encodes a lysosomal enzyme required for the hydrolysis of glycolipid glucosylceramide. Notably, homozygous *GBA* mutations are associated with a well-studied lysosomal storage disorder, Gaucher disease [37]. PD is associated with defects in PINK1–parkin-mediated mitophagy (non-mediated by mitophagy receptors) [38]. Alpha-synuclein causes the fragmentation of mitochondria, either by directly binding to them [39] or by increasing Drp1 levels [40]. Concerning PD, a promising target for maintaining mitophagy may be mitochondrial protein deubiquitinase (USP30), whose depletion in various PD models led to an improvement in mitochondrial function [5,41,42].

In turn, in cases of AD, the decrease in parkin is accompanied by an abnormal accumulation of PINK1, which is also involved in the parkin–PINK1-dependent mitophagy pathway [43,44]. The overexpression of parkin leads to the restoration of mitophagy and mitochondrial potential [44]. The impairment of lysosomal activity and defects in the lysosomal proteolysis of autophagosome contents are important events leading to AD pathogenesis. In a hereditary form of AD, it was shown that a mutation in the *PSEN1* gene encoding for transmembrane protein presenilin 1, a component of the gamma-secretase complex, leads to abnormal alkalization of lysosomes, a decrease in lysosomal hydrolase activity, and an increase in p62 levels, findings that are consistent with impaired lysosomal degradation [45]. Parkin-mediated induction of mitophagy in neurons was demonstrated both in an APP mouse AD model and in AD patients, but as the disease progresses, the level of cytosolic parkin decreases, which leads to ineffective mitophagy [45,46]. Another participant in mitophagy is sirtuin 3, which activates forkhead box, subgroup O (FOXO) for the subsequent induction of p62, and is involved in the formation of lysosomes. However, in neurons of APP/PSI mice, the expression of sirtuin 3 is reduced [47]. There is evidence from iPSC-derived AD neurons that defects in the activation of unc-51-like autophagy activating kinase 1 (ULK1) and TANK-binding kinase 1 (TBK1), related to their phosphorylation, lead to impaired initiation of mitophagy [48]. Based on these data, one can assume subsequent changes to the mitophagy pathways in the brain in AD [49,50]. Indeed, the amount of cytosolic parkin decreases, and its recruitment to the mitochondrial surface is impaired, which is probably caused by the overexpression of the protein- and tau-mediated sequestration of this parkin in the cytosol [13,50].

The role of huntingtin, the protein encoded by the *HTT* gene, has been investigated in models of Huntington’s disease. It was shown that CAG expansion in the *HTT* gene leads to the aggregation of poly(Q)-containing huntingtin and negatively affects mitophagy due to disruption of the normal function of HTT, which consists of recruiting mitophagy receptors necessary for the interaction of damaged mitochondria with autophagosomes [51]. Normally, huntingtin interacts with ULK1, playing a critical role in the initiation of autophagy [51,52]. Huntingtin dysfunction leads to the accumulation of damaged mitochondria and subsequent oxidative stress in cells [51]. The greatest damage in Huntington’s disease is observed in striatal neurons [51,52], but the reasons for this selectivity remain unclear. It was shown in the primary cultures of rat cortical and striatum neurons that mitochondrial fission is more characteristic of striatal neurons, and the inhibition of HDAC6 alpha-tubulin deacetylase in these neurons induced mitochondrial mobility and their fusion to the level observed in cortical neurons [52]. The data obtained make it possible to consider HDAC6 inhibition as a possible approach to the treatment of Huntington’s disease [52].

### 2.2. Receptor-Mediated Mitophagy in Neurons

Another mechanism is receptor-mediated mitophagy, which is initiated via mitochondrial receptors that contain LC3-interacting region (LIR) motifs (W/F/YxxL/I) and does not require PINK1 or parkin [13]. The highly selective interaction of LIR-containing proteins with specific members of ATG8 subfamily proteins mediates the sequestration of the receptor-expressing mitochondria by the growing phagophore [53]. The most studied receptors that mediate mitophagy are activating molecules in Beclin1-regulated autophagy (AMBRA1), BCL2 (interacting protein 3 (BNIP3), FUN14 domain-containing 1 (FUNDC1), and Nip3-like protein X (NIX) as well as proteins of the inner mitochondrial membrane, cardiolipin and prohibitin 2 (PHB2). Receptor-mediated mitophagy is only activated under certain conditions. For example, during hypoxia, the transcription of BNIP3 and NIX is activated upon stabilization of the hypoxia-inducible factor hypoxia-induced factor 1 (HIF1) [43]. The activity of BNIP3 and NIX is regulated by phosphorylation; their increased phosphorylation enhances their ability to bind to LC3 [44]. Hypoxia also promotes the binding of mitophagy receptor FUNDC1 to autophagosome protein LC3 to mediate mitophagy [54,55,56]. A recent study reported that NIX (but not BNIP3) is downregulated by stress-induced glucocorticoids, contributing to chronic stress-induced synaptic defects, which has been shown on mouse hippocampal neurons and human neuroblastoma SH-SY5Y cells. This decreased NIX expression is achieved through the binding of the glucocorticoid receptor to PGC-1α, which leads to its downregulation and decreased nuclear translocation, thus decreasing NIX-dependent mitophagy [57]. Figure 2 presents the scheme of receptor-mediated mitophagy. Regarding FUNDC1-dependent mitophagy, recent research shows that this pathway ensures the neuroprotective effects of treatment of ischemia–reperfusion injury. In that study, drug administration led to increased AMP-activated protein kinase (AMPK) phosphorylation, which in turn increased FUNDC1 expression [58]. 

Under normal conditions, AMBRA1 interacts with parkin, but in the absence of parkin, AMBRA1 has been shown to play the role of selective mitophagy receptor [59]. A similar situation was described for BNIP3L (Nix), ubiquitinated by parkin. At the same time, BNIP3L is able to mediate mitophagy independently of the PINK1–parkin-dependent pathway, which can play a compensatory role in PD [60,61,62]. In the nerve cells of PD patients, ULK-1, ULK-2, and AMBRA1 were shown to be present in Lewy bodies; however, expression was increased in the case of Beclin-1 in the substantia nigra of PD patients [63]. 

In the hippocampus of aging mice, a decrease in the level of AMBRA1 was observed both during normal aging and in a transgenic AD model [64], and in the latter case, it occurred at an earlier age. It should be noted that in the same work, contradictory data were obtained for the neocortex, where the content of Ambra1, Beclin1, and LC3II increased with age [64].

### 2.3. Transcellular Mitophagy of Neurons

Recent studies have described the exocytosis of mitochondria from cells, followed by their endocytosis or phagocytosis. In the brain, neurons release mitochondria at synapses, and these extracellular mitochondria are taken up by glial cells for phagocytosis [12,65]. This phenomenon is called transcellular mitophagy (Figure 3). In addition, the reverse process of mitochondrial transport from glial cells to neurons has been described [65], and this two-way transport can be regarded as an alternative method of cell-to-cell signaling. In experimental studies on models of stroke, glial cells transported mitochondria to neurons to protect the latter from hypoxia, and this process involved CD38 cells; the suppression of CD38 signaling worsened neurological outcomes [66]. The exact mechanism of transcellular mitophagy in neurons is poorly understood, but one can suggest an important role of uncoupling protein 2 (UCP2), which is involved in the regulation of reactive oxygen species [9,66]. An interesting study was conducted on human pluripotent stem cells (hPSCs) expressing a form of the Alexander disease-associated “astrocytic” gene glial fibrillary acidic protein (*GFAP*) with a mutation that impairs mitochondrial transfer from astrocytes [67].

## 3. Mitophagy in Glial Cells in Aging and Neurodegeneration

Autophagosomes are constantly forming in both neurons and astrocytes. Mitochondria occupy a significant part of the volume of thin, even, distal processes of astrocytes [68,69]. At the same time, basal mitophagy levels in astrocytes are not high, and about 1% of mitochondria co-localize with the mitophagy trigger LC3BII [68,70]. The regulation of the mitochondrial network in the processes of astrocytes is associated with the differentiation and maturation of astrocytes, their participation in the maintenance of synapses, and the glial response to damage [70]. The importance of the functioning and morphogenesis of astrocytes was shown for several proteins involved in mitochondrial biogenesis and mitophagy, including peroxisome proliferator-activated receptor gamma coactivator 1-alpha (PGC1a) [71], autophagy-related proteins ATG7 [69] and ATG5 [72], and p62 [73].

Returning to transcellular mitophagy in neurodegenerative diseases [65], it should be pointed out that impaired mitochondria can be taken up from the extracellular space by astrocytes [74]. For instance, abnormal mitochondria from dopaminergic terminals that become degenerate in PD are transported into astrocytes, where they undergo polyubiquitination and further mitophagy and, at the same time, there is activation of neuroinflammation and aggravation of the neurodegenerative process [75].

According to proteomic analysis, the pharmacological modulators of autophagy rapamycin (an inducer of mitophagy) and bafilomycin (an inhibitor of mitophagy), induce changes in the expression of numerous glial molecular targets involved in the differentiation, development, and survival of astrocytes [76]. These targets include proteins involved in the activation of astrocytes during neuroinflammation and determine their pro-inflammatory or anti-inflammatory phenotype [77]. It is often impossible to single out changes associated specifically with mitophagy since macro-autophagy and mitophagy share common regulatory mechanisms, and the number of studies devoted to mitophagy in glial cells is relatively small.

### 3.1. PINK1–Parkin Pathway of Astrocyte Mitophagy (Non-Receptor-Mediated Mitophagy)

The mechanisms of mitophagy in astrocytes have been studied in the most detail in connection with the PINK1–parkin pathway of activation. According to transcriptome analysis, PINK1 expression in human astrocytes is higher than in neurons [78,79]. It is worth noting the different levels of proteins associated with mitophagy in humans and mice, which should be taken into account when interpreting experimental results.

It has been shown that PINK1 performs important functions in glial cells, and its knockout leads to an increase in the pro-inflammatory response and disrupts astrogliogenesis and astrocyte proliferation in response to growth factors [80]. The expression of parkin under normal conditions is higher in neurons than in astrocytes; however, endoplasmic reticulum stress (unfolded protein stress) led to an increase in the expression and redistribution of parkin in astrocytes, but not in neurons [81]. Parkin knockout results in a decrease in the proliferation of astrocytes and an increase in the expression of proapoptotic proteins in astrocyte cultures [82]. The presence of special mechanisms in astrocytes that regulate the PINK1–parkin pathway of mitophagy demonstrates the fact that upon in vitro induction of mitophagy with valinomycin, PINK1 activity is increased to a greater extent in astrocytes than in neurons [25]. Interestingly, the recently characterized PINK1 antagonist, PPEF2 phosphatase, which dephosphorylates ubiquitin and suppresses mitophagy, also has greater activity in astrocytes than in neurons [83]. Another negative regulator of the parkin pathway, deubiquitinylase USP30, is active in both neurons and astrocytes, and its pharmacological suppression increases parkin-dependent mitophagy in astrocytes [84].

In general, according to the published data, it seems that astrocytes, in comparison with neurons, are characterized by more pronounced changes in the PINK1–parkin pathway of mitophagy regulation in response to various stimuli, including inflammation [68] and metabolic stress [85].

### 3.2. Receptor-Mediated Mitophagy in Glial Cells

The parkin-dependent mitophagy pathway is activated when mitochondria are damaged in response to the depolarization of their membranes. Other pathways involve mitophagy receptors located in the outer mitochondrial membrane. All of them interact with γ-aminobutyric type A (GABAA)-receptor-associated protein (ATG8/LC3/GABARAP) involved in autophagosome formation [86].

Outer membrane protein ubiquitination acts as a signal for cargo receptors p62/SQSTM1, NDP52, optineurin, and others, binding with ubiquitin and LC3B, an autophagosome membrane protein [14,87,88]. A number of studies have shown p62 participates in autophagy in astrocytes during ischemia and metabolic stress [4]; unlike astrocytes, neurons did not accumulate p62 during short-term starvation. In mitochondrial dysfunction in the context of AD, beta-amyloid accumulation, impairment of autophagy in astrocytes [49], decreased LC3-II formation, and p62 aggregation, including its co-localization with beta-amyloid peptides in astrocytes, were revealed [89]. Another mitophagy receptor, NDP52, associated with the phosphorylation of tau protein in AD, is also shown to be expressed in astrocytes surrounding amyloid plaques [90]. Although there is much data on mitophagy impairment in neurons and its relationship with tau and beta-amyloid proteinopathies [91], changes in mitophagy in astrocytes in the context of AD remain unknown.

The relationship between astrocyte pathological activation and the regulation of receptors of optineurin and p62 is evident in different neurodegenerative diseases [92]. It was shown that amyotrophic lateral sclerosis (ALS)-associated optineurin mutations cause mitophagy disorders [12]. Moreover, optineurin and p62 are phosphorylated by TBK1 kinase, which is associated with ALS [5,42]. This is based on data obtained from exome sequencing of familial ALS patients; the identification of eight loss-of-function *TBK1* mutations allowed for the conclusion that the haploinsufficiency of *TBK1* may cause ALS [93]. Optineurin in ALS is co-localized with mutant-aggregated SOD1, and these aggregates located on the outer mitochondrial membrane [69] also contain ubiquitin, LC3, and p62, which demonstrates the involvement of optineurin in autophagy [94]. It is assumed that astrocytes accumulating SOD1 have a neurotoxic effect on motor neurons [78]. Mutant astrocytes in ALS can modulate the autophagy pathway in a non-cell-autonomous manner. It was shown that the growth of cells in conditioned medium from ALS patient-derived astrocytes leads to p62 accumulation and impaired autophagy [95,96].

Interestingly, recessive mutations of the HECT domain and ankyrin repeat-containing ubiquitin ligase E3 *(HACE1*) gene, coding for a ubiquitin ligase that ubiquitinates optineurin and ensures its interaction with p62/SQSTM1, are associated with mitochondrial dysfunction and severe damage to nervous system development [97], and experimental *HACE1* knockout causes dysfunction of astrocyte mitochondria and provokes the development of astrogliosis in a model of Huntington’s disease [98]. These findings are consistent with the decreased level of this protein observed in the striatum of patients with Huntington’s disease [98].

One of the mechanisms regulating the binding of p62, NDP52, and optineurin receptors to their targets is their phosphorylation by TBK1 kinase [92]. Hemizygous deletion of the *TBK1* gene in mice does not cause neurodegeneration but leads to reactive changes in astrocytes, neuroinflammation, and p62 aggregation in neurons at a young age, which the authors associate with premature aging [99]. 

Another mitophagy receptor BNIP3 do not interact with p62 and ubiquitin and knockout of the BNIP3 gene is associated with the increased proliferation of astrocytes [100]. 

Mitophagy receptors can be characterized by different levels of expression in astrocytes and neurons; for example, activating molecule in Beclin1-regulated autophagy (AMBRA1) is practically undetectable in astrocytes [101].

### 3.3. Autophagosome-Forming Proteins

Autophagosome membrane formation is regulated by two ubiquitin-like conjugation systems, Atg12/Atg5 and LC3 complexes, and several other Atg proteins are also involved in the process [102]. The support for autophagosome formation at normal levels appears to be especially important in ontogenesis. At the initiation stage of autophagosome formation, Atg13 binds serine–threonine kinases ULK1 or ULK2 [102]. At this stage, the regulation of autophagy involves the mTOR protein kinase, which inhibits mitophagy and participates in the regulation of astrocyte functions both in normal conditions and in neurodegenerative diseases [103]. However, the details of mitophagy regulation in astrocytes via the mTOR signal pathway have not been characterized. For autophagy inducers ULK1 and ULK2, it was shown that their inhibition is associated with the development of gliomas [104], which reflects the role of autophagy regulation and the importance of maintaining normal levels for astrocyte differentiation.

In mice, the knockout of Atg5 and Atg7 (necessary for autophagosome formation) results in behavioral disturbance, neurodegeneration, and increased ROS production [52,88]. By analogy with the impairment in earlier stages of mitophagy, the dysfunction of autophagosome formation also leads to disturbances in astrocyte proliferation. For example, a deficit in Atg5 suppresses astrocyte differentiation, whereas its higher expression leads to an excessive increase in astrocyte numbers [72]. The most commonly used marker of autophagy is LC3, namely the ratio of its soluble (LC3I) to lipid-bound (LC3II) forms [105]. Cell models show that aging is characterized by higher expression of LC3 II both in neurons and astrocytes, while in starving conditions, astrocytes demonstrate increased LC3 II expression compared with neurons [105].

Compared with neurons, astrocytes appear to be able to more flexibly control the process of mitophagy in response to various influences [68,85] since their versatile participation in energy and mediator metabolism, pro-inflammatory reactions, and nerve tissue remodeling requires significant mitochondrial rearrangement [70]. In general, the dysregulation of mitophagy is characteristic of astroglia in neurodegenerative diseases, various stress effects, and aging. Most of the proteins involved in mitophagy have been shown to play an important role in astrocyte functioning [106]. Mitophagy dysfunction leads to impaired differentiation and proliferation of astroglia and its interaction with neurons [106,107]. The activation of astrocytes and their pro-inflammatory response, leading to neuronal damage, are closely related to proteins that regulate autophagy and, in particular, mitophagy.

The mechanisms of mitophagy regulation in reactive astrocytes and microglia are of particular interest in the development of therapy for neurodegenerative diseases; however, these remain poorly characterized.

In addition to astrocytes, researchers’ attention has also been drawn to the importance of mitophagy regulation in other glial cells in neurodegenerative diseases and aging. Microglia play a major role in cleaning brain tissue from neurotoxic and pathologically aggregated proteins while, at the same time, participating in regulation of the immune response [108]. As already mentioned, disturbances in the quality control of mitochondria in neurons are characteristic of AD. In addition, a decrease in the level of mitophagy in microglia was shown, which increases the level of pro-inflammatory cytokines and results in reduced phagocytosis [108,109,110]. In AD, the impairment of mitophagy is associated with the decreased phosphorylation of various proteins, including ULK1 and TBK1 [48]. It was shown, on experimental and pathological material [48], that the dysregulation of mitophagy and the accumulation of damaged mitochondria in AD not only affects neurons and astrocytes but also microglia. The restoration of the mitophagy level in AD models led to the elimination of damaged organelles and restored the phagocytic activity of microglia [107,108].

Mitophagy disorders also affect oligodendroglia. The important role of mitophagy in the differentiation of oligodendrogliocytes was demonstrated, and the regulation of mitochondrial network processes and the survival of oligodendrogliocytes is mediated by the BNIP3 receptor [110]. It is presumed that the observed white matter damage is related to defective mitophagy receptors prohibitin (PHB2) and disrupted in schizophrenia 1 (DISC1) [111,112].

## 4. Conclusions

Mitophagy pathways play an important role in maintaining physiological homeostasis, are involved in the mechanisms of aging and neurodegenerative disorders, and represent promising targets for the development of potential therapeutic agents aimed at regulating mitochondria quality control in neurons and glial cells (Table 1). A significant number of molecules that induce or inhibit mitophagy are currently under consideration [113,114], which may be useful for testing hypotheses or developing drugs for the treatment of neurodegenerative diseases. The validation of promising drugs in animal and cell models, including neurons and glial cells derived from human iPSCs as well as the elucidation of mitophagy regulation mechanisms in human samples, requires reliable biomarkers. Currently, specific biomarkers that reflect the activity of mitophagy include ubiquitin phosphorylated at serine 65, phosphorylated PINK1 and parkin, the expression and phosphorylation of several proteins of the outer mitochondrial membrane, in addition to general biomarkers of oxidative stress and neuroinflammation. At the same time, given the variety of the regulatory pathways of mitophagy, there is no doubt that this list will be expanded, eventually including indicators that reflect the state of mitophagy in certain types of cells of the nervous tissue.

## Figures and Tables

**Figure 1 ijms-22-10251-f001:**
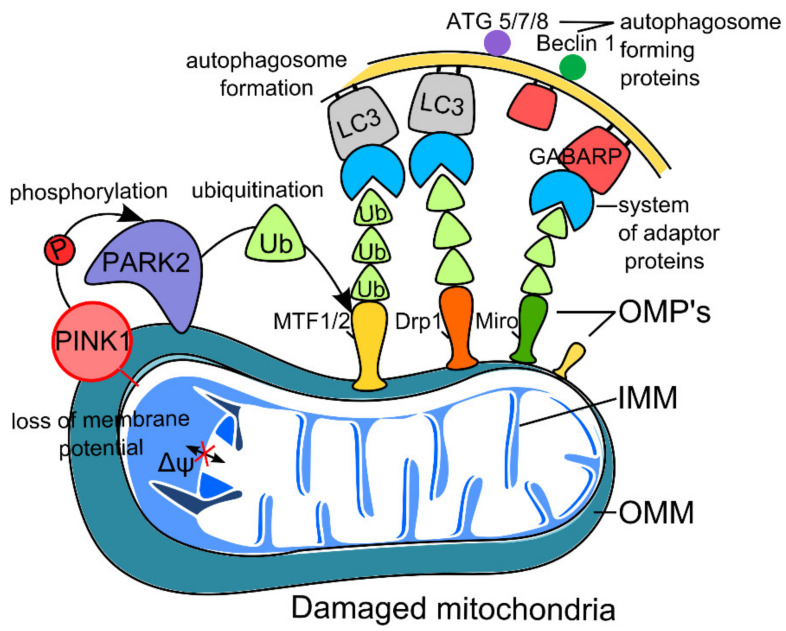
Scheme of classical non-receptor-mediated mitophagy. The first event in mitochondrial quality control within mitophagy is the detection of damaged mitochondria with loss of membrane potential. Compared with healthy mitochondria, they accumulate PTEN-induced kinase 1 (PINK 1) on the outer membrane (OMM), where it binds and activates parkin (PARK2) through its phosphorylated serine residue (S65). Then, this cascade of reactions leads to the ubiquitination of mitochondrial proteins (e.g., mitofusin 1/2, Drp1, and Miro) embedded in the outer membrane of mitochondria (OMP’s). With the help of ubiquitin-binding adapters, parkin-ubiquitinated mitochondrial substrates connect to LC3 (Microtubule-associated Protein 1A/1B-light Chain 3) and GABARAP (GABA Type A Receptor-associated Protein) proteins located on the autophagosome, followed by autophagic degradation of dysfunctional mitochondria.

**Figure 2 ijms-22-10251-f002:**
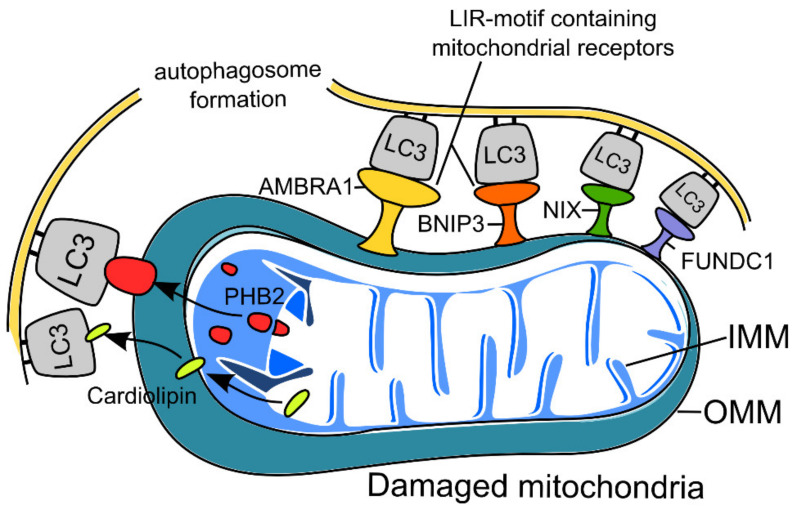
Scheme of receptor-mediated mitophagy. Receptor-mediated mitophagy is activated under specific conditions (e.g., toxins or hypoxia) via mitochondrial receptors containing LIR motifs. Among them are proteins of the outer mitochondrial membrane (OMM), e.g., AMBRA1 (Activating Molecule in Beclin1-regulated Autophagy), BNIP3 (BCL2 Interacting Protein 3), FUNDC1 (FUN14 Domain Containing 1), and NIX (Nip3-like protein X), as well as proteins of the inner mitochondrial membrane (IMM), cardiolipin, and PHB2 (Prohibitin 2). Subsequently, mitophagy receptors bind with autophagosome protein LC3 to mediate mitophagy.

**Figure 3 ijms-22-10251-f003:**
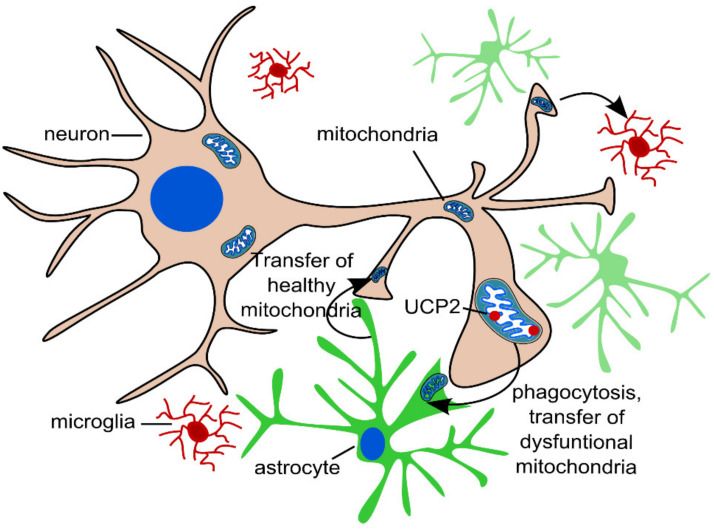
Scheme of transcellular mitophagy. Damaged mitochondria undergo exocytosis in synapses. Then, extracellular organelles can be removed by neighboring glial cells through endocytosis or phagocytosis. Uncoupling protein 2 (UCP2) could play an important role in this process. The reverse process of mitochondrial transport from glial cells to neurons has been also described.

**Table 1 ijms-22-10251-t001:** Potential biomarkers of mitophagy disruption in different types of cells of the nervous tissue during aging and age-associated diseases.

	Protein Abbreviation	Full Name	AD	PD	HT	ALS	Aging	Citation
**Non-receptor-** **mediated mitophagy**	PINK1	Phosphatase and tensin homolog (PTEN)-induced kinase	+	+			+	[12,13,23,24,25,38,68,85]
	USP30	Ubiquitin carboxyl-terminal hydrolase 30		+				[5,41,42,84]
	ULK1	Unc-51-like autophagy activating kinase 1	+		+			[48,51,52,93,102,104]
	TBK1	TANK-binding kinase 1	+			+		[5,42,48,92]
	Drp1	Dynamin-related protein 1 (Drp1)		+			+	[40]
	p62/SQSTM1	Sequestosome 1	+			+	+	[14,73,87,88,96,99]
	NDP52	Nuclear dot protein 52 kDa	+					[90,92]
	Optn	Optineurin	+			+		[48,94]
**Receptor-mediated mitophagy**	FUNDC1	FUN14 domain-containing protein 1 (phosphorylation)	+					[48,54,55,56]
	AMBRA1	Autophagy and Beclin 1 regulator 1	+				+	[43,101]
	BNIP3	BCL2-interacting protein 3 (phosphorylation)	+					[96]
**Autophagosome-forming proteins**	GABARP	γ-aminobutyric-acid type A receptor-associated proteins		+				[12,14,86]
	LC3-II	Microtubule-associated protein 1A/1B-light chain	+	+			+	[14,87,88,89,102,105]
	BECN1	Beclin-1	+				+	[44]
	ATG5/ATG7	Autophagy-related genes, associated with LC3 modification		+				[68,72,102,104]

## Data Availability

Not applicable.

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
