# Peer review of "Impaired Mitophagy in Neurons and Glial Cells during Aging and Age-Related Disorders"

_ijms, 2021, doi:10.3390/ijms221910251_

Round 1

Reviewer 1 Report

The topic is not new. You can find similar papers just by a simple search. For instance, https://pubmed.ncbi.nlm.nih.gov/31050206/. 

The English language needs deep revision by a native English speaker who is familiar with academic writing.

I'm surprised to see a review paper with just one table and one chart. The comprehensive paper should contain more descriptive figures to illustrate the main concept of the paper.

Author Response

Dear Reviewer!

Thank you for thoroughly reviewing our manuscript “Disruption of mitophagy in different cells of the nervous tissue during aging and age-associated diseases: prospects for the development of new biomarkers” submitted to International Journal of Molecular Sciences (Special Issue «Molecular mechanisms and pharmacological targeting of neuroprotection»).

We appreciate the time and effort you put in to provide feedback on our article, and we are grateful for the insightful comments on and valuable improvements to the manuscript. We have incorporated the changes to the text according to your suggestions and highlighted them throughout the manuscript.

Please see below point-by-point response to your comments and concerns.

1. The topic is not new. You can find similar papers just by a simple search. For instance, https://pubmed.ncbi.nlm.nih.gov/31050206/. 

We agree that this topic is very "hot" in modern neuroscience, but nevertheless we believe that our article makes a valuable contribution to the field. Over the past several years, it has become evident that various hallmarks of brain aging at the cellular level are closely related to mitophagy. Accordingly, the number of publications covering this problem from new and new positions is constantly growing. To present and analytically summarize the most up-to-date data, we reviewed in the article more than 30 publications from 2020-2021 (and several new publications from 2021 were included in this updated version of the manuscript). The mechanisms of mitophagy activation in different types of cells of the nervous are very numerous and heterogeneous, and our review discusses mitophagy both in neurons and in glial cells, in contrast to the majority of other articles focused predominantly on neurons.

2. The English language needs deep revision by a native English speaker who is familiar with academic writing.

We used English Editing Service of International Journal of Molecular Sciences provided by Specialist Edit.

3. I'm surprised to see a review paper with just one table and one chart. The comprehensive paper should contain more descriptive figures to illustrate the main concept of the paper.

Thank you for pointing this out. We provided 3 additional figures according to the three mechanisms of mitophagy. Also we divided the manuscript into different consecutive sections and changed Table 1.

Reviewer 2 Report

This article titled ‘Disruption of Mitophagy in different cells of the nervous tissue during aging and age-associated diseases” Prospects for the development of new biomarkers’ by Sukhorukove V. et al. is an attempt to identify the effects of mitophagy as a common target of ageing and neurodegeneration.

In my opinion, present study is interesting. However, some minor corrections are required.

  1. Please correct English in this manuscript.
  2. In title, it is better to revise ‘different cells’ to ‘different types of cells’ of the nervous tissue.
  3. Please write full spelling before the first mentioned abbreviation (ATG, PINK1, DJ-1..)  
  4. Please use the same term. For example, ‘non-receptor mediated mitophagy’ vs ‘receptor-non-mediated mitophagy’ ‘ageing’ vs ‘aging’.
  5. Write the full spelling of the abbreviation in figure legends.
  6. Please add references in line number 75 (Mitophagy in neurons~ and transcellular mitophagy)

Author Response

Dear Reviewer!

Thank you for thoroughly reviewing our manuscript “Disruption of mitophagy in different cells of the nervous tissue during aging and age-associated diseases: prospects for the development of new biomarkers” submitted to International Journal of Molecular Sciences (Special Issue «Molecular mechanisms and pharmacological targeting of neuroprotection»).

We appreciate the time and effort you put in to provide feedback on our article, and we are grateful for the insightful comments on and valuable improvements to the manuscript. We have incorporated the changes to the text according to your suggestions and highlighted them throughout the manuscript.

Please see below point-by-point response to your comments and concerns. All page numbers refer to the revised manuscript file with tracked changes.

1. Please correct English in this manuscript.

We used English Editing Service of International Journal of Molecular Sciences provided by Specialist Edit.

2. In title, it is better to revise ‘different cells’ to ‘different types of cells’ of the nervous tissue.

Thank you for this suggestion. We revised the title of the article according your comment and suggestions of other Reviewers. We decided that the title “Impact of Impaired Mitophagy on Neurons and Glial Cells during Aging and Age-related disorders” would be more correct.

3. Please write full spelling before the first mentioned abbreviation (ATG, PINK1, DJ-1…)  

We have added the full spelling before the first mentioned abbreviation on P2L57-59, P7L287-289.

4. Please use the same term. For example, ‘non-receptor mediated mitophagy’ vs ‘receptor-non-mediated mitophagy’ ‘ageing’ vs ‘aging’.

Thank you for pointing this out. These mistakes have been corrected.

5. Write the full spelling of the abbreviation in figure legends.

Thank you for this suggestion. We decided to replace the previous scheme with several descriptive figures, and the full spelling of the abbreviations is provided in text next to figures.

6. Please add references in line number 75 (Mitophagy in neurons~ and transcellular mitophagy)

Thank you for pointing this out. The references have been added on P2L78-80.

Reviewer 3 Report

The manuscript reviews the role of mitophagy in neuronal/glial homeostasis in aging and neurodegenerative disorders, such as Parkinson’s disease (PD) and Alzheimer’s disease (AD). In particular, they focused on mitophagy pathways that provide a number of promising targets for potential therapeutic agents.

The review is interesting but highlight some critical aspects:

They introduced three mechanisms of mitophagy including receptor-non-mediated mitophagy, receptor-mediated mitophagy, and transcellular mitophagy in section 2. In this regard, they should add the description of transcellular mitophagy in figure 1. Further, they should standardize the different sections of the manuscript also according to three mechanisms of mitophagy including the table 1. This can help the reader to understand the role of the different proteins in the mitophagy mechanisms.

The age is a risk factor for both PD and AD and it may share common pathways of mitophagy with neurodegenerative processes. In this regard, they should highlight the common proteins involved in the mitophagy mechanisms in both aging and neurodegenerative diseases with a new section and/or table.

The authors emphasize the term “Biomarkers” in the title. However, they investigated this aspect in a marginal manner in introduction and conclusion section. In addition, they did not consider this term in the abstract. They should therefore improve and deepen the description of this aspect in introduction and conclusion section.

They should shorten and improve the title of the manuscript. Example, “Disruption” and “in Different cells” are out of tune.

Author Response

Dear Reviewer!

Thank you for thoroughly reviewing our manuscript “Disruption of mitophagy in different cells of the nervous tissue during aging and age-associated diseases: prospects for the development of new biomarkers” submitted to International Journal of Molecular Sciences (Special Issue «Molecular mechanisms and pharmacological targeting of neuroprotection»).

We appreciate the time and effort you put in to provide feedback on our article, and we are grateful for the insightful comments on and valuable improvements to the manuscript. We have incorporated the changes to the text according to your suggestions and highlighted them throughout the manuscript.

Please see below point-by-point response to your comments and concerns.

  1. They introduced three mechanisms of mitophagy including receptor-non-mediated mitophagy, receptor-mediated mitophagy, and transcellular mitophagy in section 2. In this regard, they should add the description of transcellular mitophagy in figure 1. Further, they should standardize the different sections of the manuscript also according to three mechanisms of mitophagy including the table 1. This can help the reader to understand the role of the different proteins in the mitophagy mechanisms.

We think this is an excellent suggestion. We decided to replace the previous scheme with several descriptive figures according to the three mechanisms of mitophagy. Also we divided the manuscript into the different sections and changed Table 1 according to described mechanisms of mitophagy.

  1. The age is a risk factor for both PD and AD and it may share common pathways of mitophagy with neurodegenerative processes. In this regard, they should highlight the common proteins involved in the mitophagy mechanisms in both aging and neurodegenerative diseases with a new section and/or table.

We tried to redesign Table 1 according to your suggestions (we divided section “Age- associated diseases” into separate columns and added additional column “aging” to highlight the common proteins in both aging and neurodegenerative diseases).

  1. The authors emphasize the term “Biomarkers” in the title. However, they investigated this aspect in a marginal manner in introduction and conclusion section. In addition, they did not consider this term in the abstract. They should therefore improve and deepen the description of this aspect in introduction and conclusion section.

and

  1. They should shorten and improve the title of the manuscript. Example, “Disruption” and “in Different cells” are out of tune.

Thank you for pointing this out. Following your comments we decided that the title “Impact of Impaired Mitophagy on Neurons and Glial Cells during Aging and Age-related disorders” would be more consistent with the article content.

Round 2

Reviewer 1 Report

The title is liked the original reserach, not a review paper. Please edit it. 

Figures should be created more scientifically.

Author Response

Dear Reviewer!

Thank you for thoroughly reviewing our manuscript. Please see below point-by-point response to your comments and concerns.

  1. The title is liked the original reserach, not a review paper. Please edit it.

We agree with your assessment . The revised title reads as follows on ''Impaired Mitophagy in Neurons and Glial Cells During Aging and Age-related Disorders''. 

     2. Figures should be created more scientifically

As suggested by you, we revised our figures throughout the manuscript.